# SARS-CoV-2 Antigen Test Results to Infer Active or Non-Active Virus Replication Status in COVID-19 Patients

**DOI:** 10.3390/diagnostics12061338

**Published:** 2022-05-28

**Authors:** Giulia De Angelis, Giulia Menchinelli, Flora Marzia Liotti, Simona Marchetti, Alessandro Salustri, Antonietta Vella, Rosaria Santangelo, Brunella Posteraro, Maurizio Sanguinetti

**Affiliations:** 1Dipartimento di Scienze Biotecnologiche di Base, Cliniche Intensivologiche e Perioperatorie, Università Cattolica del Sacro Cuore, 00168 Rome, Italy; giulia.deangelis@unicatt.it (G.D.A.); giulia.menchinellli@hotmail.it (G.M.); floramarzialiotti@gmail.com (F.M.L.); alessandrosalustri032@gmail.com (A.S.); rosaria.santangelo@policlinicogemelli.it (R.S.); maurizio.sanguinetti@unicatt.it (M.S.); 2Dipartimento di Scienze di Laboratorio e Infettivologiche, Fondazione Policlinico Universitario A. Gemelli IRCCS, 00168 Rome, Italy; simona.marchetti@policlinicogemelli.it (S.M.); antonietta.vella@policlinicogemelli.it (A.V.); 3Dipartimento di Scienze Mediche e Chirurgiche, Fondazione Policlinico Universitario A. Gemelli IRCCS, 00168 Rome, Italy

**Keywords:** antigen detection, COVID-19, reverse transcription-PCR, SARS-CoV-2, subgenomic RNA, virus replication

## Abstract

We used nasopharyngeal swab samples of patients with a symptomatic (n = 82) or asymptomatic (n = 20) coronavirus disease 2019 (COVID-19) diagnosis to assess the ability of antigen detection tests to infer active (potentially transmissible) or inactive (potentially non-transmissible) infection by severe acute respiratory syndrome coronavirus 2 (SARS-CoV-2). Using the subgenomic RNA (sgRNA) as an active replication marker of SARS-CoV-2, 48 (76.2%), 56 (88.9%), and 63 (100%) of 63 samples with sgRNA positive results tested positive with the SD BIOSENSOR STANDARD Q COVID-19 Ag (Standard Q), the SD BIOSENSOR STANDARD F COVID-19 Ag FIA (Standard F), or the Fujirebio LUMIPULSE G SARS-CoV-2 Ag (Lumipulse) assay, respectively. Conversely, 37 (94.9%), 29 (74.4%), and 7 (17.9%) of 39 samples with sgRNA negative results tested negative with Standard Q, Standard F, or Lumipulse, respectively. Stratifying results by the number of days of symptoms before testing revealed that most antigen positive/sgRNA positive results were among samples tested at 2–7 days regardless of the assay used. Conversely, most antigen negative/sgRNA negative results were among samples tested at 16–30 days only when Standard Q or Standard F were used. In conclusion, based on our findings, a negative antigen test, especially with the Lumipulse assay, or a positive antigen test, especially with the Standard F assay, may suggest, respectively, the absence or presence of replication-competent SARS-CoV-2.

## 1. Introduction

Reverse transcription-PCR (RT-PCR)-based molecular assays offer a reliable means of detecting the severe acute respiratory syndrome coronavirus 2 (SARS-CoV-2) genomic RNA (gRNA) in respiratory tract samples from patients who develop coronavirus disease 2019 (COVID-19) within 1 to 2 weeks of SARS-CoV-2 exposure [1]. Whereas there is no established RT-PCR cycle threshold (Ct) value at which individuals are likely to no longer be infectious [2], growing evidence shows that RT-PCR detection of SARS-CoV-2 subgenomic RNA (sgRNA), as with viral culture (i.e., the currently accepted reference standard for virus replication) [3], may be reliably used to measure infectivity [4,5,6]. Unlike diagnostic (gRNA-detecting) RT-PCR, detection of the SARS-CoV-2 antigen, which is per se an indicator of active infection, offers an accurate but less sensitive means of identifying acutely infected individuals [7] and, therefore, the possibility to identify individuals who are shedding the infectious virus and are likely to transmit SARS-CoV-2 [2]. Compared to RT-PCR assays, which have widely replaced viral culture to diagnose respiratory infections [2], antigen assays have 100% specificity, whereas their sensitivity may be moderately high (84%) or low (53%) for symptomatic (i.e., tested within the first 7 days of illness) or asymptomatic patients, respectively [8].

Only one study so far [9], to the best of our knowledge, has compared SARS-CoV-2 sgRNA with SARS-CoV-2 antigen detection in symptomatic COVID-19 patients, showing that antigen was concordant with sgRNA for samples from a midturbinate nasal swab but not from a nasopharyngeal swab (NPS). To clarify this matter, we used NPS samples of asymptomatic or symptomatic COVID-19 patients to assess the antigen detection tests’ ability for differentiating between active (potentially transmissible) and inactive (potentially non-transmissible) SARS-CoV-2 infection. We assessed the SD BIOSENSOR STANDARD Q COVID-19 Ag, the SD BIOSENSOR STANDARD F COVID-19 Ag FIA, and the Fujirebio LUMIPULSE G SARS-CoV-2 Ag in comparison with SARS-CoV-2 sgRNA, here used as a marker of active virus replication.

## 2. Materials and Methods

**Study design and clinical samples.** This retrospective study was carried out at the Fondazione Policlinico Universitario A. Gemelli IRCCS hospital of Rome, Italy, and approved by the Institutional Ethics Committee (reference no. 0040205/21), during a 2021 one-month (24 November through 24 December) period. We randomly selected NPS samples from patients who had a laboratory-confirmed COVID-19 diagnosis, which relied on a positive RT-PCR result through a Panther Fusion SARS-CoV-2 assay (Hologic S.r.l., Roma, Italy). Residual 2-mL aliquots from originally collected samples in 3 mL of universal transport medium (UTM, Copan, Brescia, Italy) were put into anonymized blank vials and immediately processed for or kept at −80 °C until testing (see below). One hundred and two NPS samples from patients diagnosed with asymptomatic (n = 20), mild (n = 14), moderate (n = 17), severe (n = 32), or critical (n = 19) COVID-19, according to a median (interquartile range) RT-PCR Ct value of 24.6 (18.6–30.2) for SARS-CoV-2 gRNA detection, were included (Table 1). Only for symptomatic patients (n = 82), disease severity was established according to the criteria from US National Institutes of Health (https://www.covid19treatmentguidelines.nih.gov accessed on 10 May 2022).

**SARS-CoV-2 RNA and antigen testing.** Following diagnostic RT-PCR (i.e., using a molecular assay not allowing Ct value determination as above specified), samples (i.e., residual NPS samples) were used to detect SARS-CoV-2 gRNA (i.e., using a molecular assay allowing Ct value determination), sgRNA, and antigen. For gRNA, Ct values for three SARS-CoV-2 targets, such as envelope (E), RNA-dependent RNA polymerase (RdRP)/spike (S), and nucleoprotein (N) encoding genes, were determined using an updated version of the Allplex™ 2019-nCoV assay [10], now designated as Allplex™ SARS-CoV-2 assay. For sgRNA (i.e., E-gene sgRNA) detection, we used an in-house RT-PCR assay [11], which was essentially in accordance with a method described elsewhere [4]. For antigen detection, we used STANDARD Q COVID-19 Ag (SD BIOSENSOR, https://www.sdbiosensor.com accessed on 10 May 2022), STANDARD F COVID-19 Ag FIA (SD BIOSENSOR, https://www.sdbiosensor.com accessed on 10 May 2022), and LUMIPULSE G SARS-CoV-2 Ag (Fujirebio, https://www.fujirebio.com accessed on 10 May 2022) immunoassays (hereafter referred to as Standard Q, Standard F, and Lumipulse, respectively) according to the respective manufacturer’s instructions. Briefly, the Standard Q is a lateral flow assay that detects SARS-CoV-2 N antigen in a cassette-based format with a visual read-out, allowing interpretation of results as positive (i.e., test band detected) or negative (i.e., test band not detected). The Standard F is a fluorescence immunoassay that detects SARS-CoV-2 N antigen with STANDARD F Analyzer, allowing interpretation of results as positive (i.e., cutoff index (COI) value of ≥1.0) or negative (i.e., COI of <1.0), according to the manufacturer’s instructions (https://www.sdbiosensor.com accessed on 10 May 2022). The Lumipulse uses a CLEIA (chemiluminescent enzyme immunoassay) technology to measure SARS-CoV-2 S antigen by a LUMIPULSE^®^ G System automated instrument, allowing interpretation of results as positive (>10–>5000 pg/mL), gray-zone positive (1.34–10 pg/mL), or negative (<1.34 pg/mL) [12].

**Statistical analysis.** For SARS-CoV-2 antigen or sgRNA assays’ results, differences between a priori established groups were assessed using the chi-square test or the Wilcoxon test, as appropriate. Percent agreement values, with their respective confidence intervals (CIs), were calculated comparing results from antigen assays with those from the sgRNA assay, which was used as the reference method. Statistical analysis was conducted using Stata 15 (StataCorp, College Station, TX, USA) or GraphPad Prism 7 (GraphPad Software, San Diego, CA, USA) software. *p* < 0.05 was considered statistically significant.

## 3. Results

For the 102 patients studied, ages ranged from 45.7 to 78.8 years (median 62.8), and 53 (52.0%) out of 102 were male (Table 1). Excluding asymptomatic patients (n = 20), time from the onset of symptoms to sample collection ranged from 1 day to >30 days, with most patients (32/82; 39.0%) being sampled 2–7 days after symptoms became apparent (Table 1). Accordingly, in this subset of patients, Standard Q, Standard F, or Lumipulse antigen positivity rates were found to be the highest (28/32 [87.5%], 29/32 [90.6%], and 32/32 [100%], respectively). Lower rates were, indeed, noticed in subsets of patients sampled at 8–15 days (7/28 (25.0%), 18/28 (64.3%), and 28/28 (100%), respectively) or 16–30 days (0/13 (0.0%), 2/13 (15.4%), and 9/13 (69.2%), respectively) after the onset of symptoms, which accounted for 28 (34.1%) and 13 (15.8%) of the 82 samples collected in total.

Of the 102 samples studied (Table 1), 95 (93.1%) tested positive with Lumipulse, with 50 (49.0%) and 66 (64.7%) found to be concomitantly positive with the Standard Q or the Standard F, respectively. In parallel, 63 (61.8%) samples tested positive with the sgRNA detection assay, and their median (interquartile range) RT-PCR Ct value was 33.0 (28.4–36.4). As with diagnostic RT-PCR Ct values (*p* value < 0.001 for all comparisons), sgRNA RT-PCR Ct values differed significantly between antigen positive and antigen negative samples’ groups (*p* values < 0.001 and 0.008 for Standard Q’s or Standard F’s comparisons, respectively). Seven (6.9%) samples with Lumipulse antigen (and sgRNA) negative results had a diagnostic RT-PCR Ct median value of 35.1, which was higher than the median value of 32.4 noticed in 39 (38.2%) samples with sgRNA negative results. Except for seven samples, detectable sgRNA levels were noticed in the group of 52 (51.0%) samples with Standard Q antigen negative results (RT-PCR Ct median value, 37.4) or the group of 36 (35.3%) samples with Standard F antigen negative results (RT-PCR Ct median value, 37.0).

Compared to sgRNA results (Table 2 and Figure 1), 48 (76.2%), 56 (88.9%), and 63 (100%) of 63 samples with sgRNA positive results tested positive with Standard Q, Standard F, or Lumipulse, respectively. Conversely, 37 (94.9%), 29 (74.4%), and 7 (17.9%) out of 39 samples with sgRNA negative results tested negative with Standard Q, Standard F, or Lumipulse, respectively

**Table 1 diagnostics-12-01338-t001:** Demographic and clinical characteristics of COVID-19 patients who had NPS samples analyzed for SARS-CoV-2 antigen and sgRNA ^a^.

		Standard Q			Standard F			Lumipulse			sgRNA		
Characteristic	Total	Positive	Negative	*p*-Value	Positive	Negative	*p*-Value	Positive	Negative	*p*-Value	Positive	Negative	*p*-Value
Total no. (%)	102 (100)	50 (49.0)	52 (51.0)		66 (64.7)	36 (35.3)		95 (93.1)	7 (6.9)		63 (61.8)	39 (38.2)	
Patients, n = 102													
Age	62.8 (45.7–78.8)	64.4 (52.8–79.1)	61.5 (42.5–75.1)	0.18	64.5 (52.8–79.5)	60.4 (38.3–68.4)	0.01	62.7 (45.7–78.8)	64.7 (28.9–80.8)	0.66	61.5 (49.1–76.0)	64.7 (42.7–80.2)	0.85
Male sex	53 (52.0)	23 (46.0)	30 (57.7)	0.24	34 (51.5)	19 (52.8)	0.90	49 (51.6)	4 (57.1)	0.77	29 (46.0)	24 (61.5)	0.13
Type of illness													
Asymptomatic	20 (19.6)	8 (16.0)	12 (23.1)	0.37	10 (15.2)	10 (27.7)	0.13	18 (19.0)	2 (28.6)	0.54	12 (19.1)	8 (20.5)	0.85
Mild	14 (13.7)	5 (10.0)	9 (17.3)	0.28	5 (7.5)	9 (25.0)	0.01	12 (12.6)	2 (28.6)	0.24	6 (9.5)	8 (20.5)	0.11
Moderate	17 (16.7)	6 (12.0)	11 (21.1)	0.21	12 (18.2)	5 (13.9)	0.58	17 (17.9)	0 (0.0)	0.22	10 (15.9)	7 (18.0)	0.78
Severe	32 (31.4)	19 (38.0)	13 (25.0)	0.15	26 (39.4)	6 (16.7)	0.01	31 (32.6)	1 (14.2)	0.31	21 (33.3)	11 (28.2)	0.58
Critical	19 (18.6)	12 (24.0)	7 (13.5)	0.17	13 (19.7)	6 (16.7)	0.70	17 (17.9)	2 (28.6)	0.48	14 (22.2)	5 (12.8)	0.23
Samples, n = 102													
Diagnostic RT-PCR Ct	24.6 (18.6–30.2)	18.6 (16.3–21.0)	30.1 (27.1–33.6)	<0.001	19.6 (17.1–24.4)	32.2 (29.3–35.0)	<0.001	24.1 (18.5–29.2)	35.1 (32.8–35.5)	<0.001	19.5 (17.0–24.1)	32.4 (28.9–35.1)	<0.001
sgRNA RT-PCR Ct ^b^	33.0 (28.4–36.4)	30.5 (28.0–34.1)	37.4 (33.0–38.9)	<0.001	31.9 (28.0–36.1)	37.0 (35.0–39.9)	0.008	33.0 (28.4–36.4)			33.0 (28.4–36.4)		
Days after onset of symptoms when sample was collected, n = 82													
1	7 (6.9)	6 (12.0)	1 (1.9)	0.04	6 (9.1)	1 (2.8)	0.23	6 (6.3)	1 (14.3)	0.42	5 (7.9)	2 (5.1)	0.58
2–7	32 (31.4)	28 (56.0)	4 (7.7)	<0.001	29 (43.9)	3 (8.3)	<0.001	32 (33.7)	0 (0.0)	0.06	30 (47.6)	2 (5.1)	<0.001
8–15	28 (27.4)	7 (14.0)	21 (40.4)	0.003	18 (27.3)	10 (27.8)	0.96	28 (29.5)	0 (0.0)	0.09	14 (22.2)	14 (35.9)	0.13
16–30	13 (12.7)	0 (0.0)	13 (25.0)	<0.001	2 (3.0)	11 (30.6)	<0.001	9 (9.5)	4 (57.1)	<0.001	1 (1.6)	12 (30.8)	<0.001
>30	2 (2.0)	1 (2.0)	1 (1.9)	0.97	1 (1.5)	1 (2.8)	0.66	2 (2.1)	0 (0.0)	0.69	1 (1.6)	1 (2.6)	0.73

^a^ Values are no. (%) or median (interquartile range). COVID-19, coronavirus disease 2019; Ct, cycle threshold; NPS, nasopharyngeal swab; RT-PCR, reverse transcription-PCR; SARS-CoV-2, severe acute respiratory syndrome coronavirus 2; sgRNA, subgenomic RNA. A *p*-value (positive vs negative) of ≤0.05 was considered statistically significant. ^b^ Available only for samples with sgRNA levels detected (n = 63, in total), which were distributed among antigen positive or negative testing groups (as detailed in Table 2). All samples in the Lumipulse antigen negative group (n = 7) had sgRNA levels not detected.

**Table 2 diagnostics-12-01338-t002:** Comparison of SARS-CoV-2 antigen and sgRNA results for NPS samples tested at indicated day(s) from symptom onset ^a^.

Day(s) of Symptoms to Antigen Testing	No. Samples with Results					Agreement between Antigen and sgRNA Results (95% Confidence Interval)	
	Total	Antigen Positive/ sgRNA Positive	Antigen Negative/ sgRNA Negative	Antigen Positive/ sgRNA Negative	Antigen Negative/ sgRNA Positive	Positive Percent Agreement	Negative Percent Agreement
Standard Q (all samples)	102	48	37	2	15	76.2 (63.8–86.0)	94.9 (82.7–99.4)
Asymptomatic	20	8	8	0	4	66.7 (34.9–90.1)	100.0 (63.1–100)
1 day	7	5	1	1	0	100 (47.8–100)	50.0 (1.3–98.7)
2–7 days	32	28	2	0	2	93.3 (77.9–99.2)	100 (15.8–100)
8–15 days	28	6	13	1	8	42.9 (17.7–71.1)	92.9 (66.1–99.8)
16–30 days	13	0	12	0	1	0.0 (0.0–97.5)	100 (73.0–100)
>30 days	2	1	1	0	0	100 (2.5–100)	100 (2.5–100)
Standard F (all samples)	102	56	29	10	7	88.9 (78.4–95.4)	74.4 (57.9–87.0)
Asymptomatic	20	10	8	0	2	83.3 (51.6–97.9)	100 (63.1–100)
1 day	7	5	1	1	0	100 (47.8–100)	50.0 (1.3–98.7)
2–7 days	32	29	2	0	1	96.7 (82.8–99.9)	100 (15.8–100)
8–15 days	28	11	7	7	3	78.6 (49.2–95.3)	50.0 (23.0–77.0)
16–30 days	13	0	10	2	1	0.0 (0.0–97.5)	83.3 (51.6–97.9)
>30 days	2	1	1	0	0	100 (2.5–100)	100 (2.5–100)
Lumipulse (all samples)	102	63	7	32	0	100 (94.3–100)	17.9 (7.5–33.5)
Asymptomatic	20	12	2	6	0	100 (73.5–100)	25.0 (3.2–65.1)
1 day	7	5	1	1	0	100 (47.8–100)	50.0 (1.3–98.7)
2–7 days	32	30	0	2	0	100 (88.4–100)	0.0 (0.0–84.2)
8–15 days	28	14	0	14	0	100 (76.8–100)	0.0 (0.0–23.29)
16–30 days	13	1	4	8	0	100 (2.5–100)	33.3 (9.9–65.1)
>30 days	2	1	0	1	0	100 (2.5–100)	0.0 (0.0–97.5)

^a^ Twenty of 102 samples were from asymptomatic patients at the time of COVID-19 diagnosis, which relied on reverse transcription-PCR genomic RNA detection. COVID-19, coronavirus disease 2019; NPS, nasopharyngeal swab; SARS-CoV-2, severe acute respiratory syndrome coronavirus 2; sgRNA, subgenomic RNA.

**Figure 1 diagnostics-12-01338-f001:**
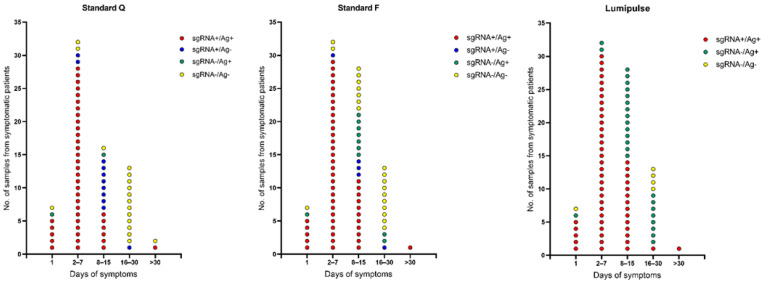
Distribution of gRNA or sgRNA reverse transcription-PCR Ct values for NPS samples from 102 COVID-19 patients tested for antigen detection with Standard Q, Standard F, or Lumipulse. Colors indicate groups of results stratified as sgRNA positive/Ag positive (red), sgRNA positive/Ag negative (blue), sgRNA negative/Ag positive (green), or sgRNA negative/Ag negative (yellow). Ag, antigen; Ct, cycle threshold; COVID-19, coronavirus disease 2019; gRNA, genomic RNA; NPS, nasopharyngeal swab; sgRNA, subgenomic RNA.

Stratifying antigen results by number of days of symptoms before sample collection (Table 2 and Figure 2) showed that Standard F for samples tested at 2–7 days had positive and negative percent agreement values of 96.7% and 100%, which in combination were found to be better than those of Standard Q (93.3% and 100%) or Lumipulse (100% and 0.0%). As illustrated in Figure 2, antigen positive/sgRNA positive results were most frequent among the samples tested at 2–7 days regardless of the (Standard Q, Standard F, or Lumipulse) test used, and antigen negative/sgRNA negative results were most frequent among the samples tested at 16–30 days only when Standard Q or Standard F were used.

## 4. Discussion

We comparatively assessed SARS-CoV-2 antigen detection tests, i.e., three among a myriad of commercially available tests [13], whose analytical sensitivity (Lumipulse > Standard F > Standard Q) depends on how advanced the immunoassay’s technology is (lateral flow assay < immunofluorescence < chemiluminescence). Two are rapid antigen tests [13] and the third (Lumipulse) is a laboratory-based antigen test [12]. Our findings show that a negative antigen test, especially with the Lumipulse assay, or a positive antigen test, especially with the Standard F assay, may be highly suggestive of the absence or presence of replication-competent SARS-CoV-2, respectively. Differentiation between the persistent shedding of viral RNA (i.e., non-infectious status), which is usually associated with a positive SARS-CoV-2 molecular assay [7], and active virus replication (i.e., infectious status) is essential for the clinical management, discontinuation of isolation, and/or work reintegration of SARS-CoV-2-infected individuals. A negative antigen result may be particularly important in the context of a previously positive SARS-CoV-2 molecular assay, i.e., within 10–14 days following symptom onset when both the detection of the infectious virus and contagiousness (or transmissibility) are almost unlikely [14].

Previous studies’ findings supported us to adopt SARS-CoV-2 sgRNA RT-PCR as the reference method for our antigen tests’ assessment [6,15,16,17]. One study [15] using respiratory tract samples obtained from COVID-19 patients within 7 days of symptom onset showed the correlation between the SARS-CoV-2 antigen (using the Becton Dickinson BD Veritor System for Rapid Detection of SARS-CoV-2 test) and SARS-CoV-2 culture (using a VeroE6TMPRSS2 cell line-based high-sensitive culture test) positivity. Another study [16] showed that three lateral flow assays and a (microfluidic) immunofluorescence assay correlated with cell-culture infectivity more significantly than SARS-CoV-2 gRNA RT-PCR. Finally, in Santos Bravo et al.’s study [6], sgRNA RT-PCR proved to detect replication-competent SARS-CoV-2 in COVID-19 patients’ NPS or other respiratory samples with a sensitivity of 97% and a positive predictive value of 94% compared to SARS-CoV-2 culture. However, to comply with the assumption that culture may be suboptimal to detect the presence of the infectious virus due to poor sensitivity [2], we used a highly sensitive RT-PCR method (i.e., the sgRNA detection assay) to assess antigen assays in our study. Similar to us, Santos Bravo et al. [6] chose to measure E-gene sgRNA, i.e., the sgRNA species known as the best option to detect infectivity so far [2].

Our study has some intrinsic limitations. First, the sample size is relatively small, particularly for patients without COVID-19 compatible symptoms who represent only 19.6% of the patients included in the study. This precludes our results from being informative for study populations that primarily consist of asymptomatic patients. Second, antigen assays studied by us are very different platforms, implying their analytical sensitivities are bound to be different. This was especially noted for asymptomatic patients, but it could also explain the high rates of Standard Q or Standard F disagreement with the sgRNA RT-PCR assay that occurred for samples tested after 7 days or more of symptoms. Third, our results suggest that antigen assays can be very helpful at the onset of symptoms, which is pretty much expected.

In conclusion, our findings reinforce the potential of SARS-CoV-2 antigen tests to serve as a marker for active SARS-CoV-2 replication. However, further studies are required before using such tests to predict SARS-CoV-2 infectivity and, thus, inform the discontinuation of COVID-19 transmission-based precautions.

## Figures and Tables

**Figure 2 diagnostics-12-01338-f002:**
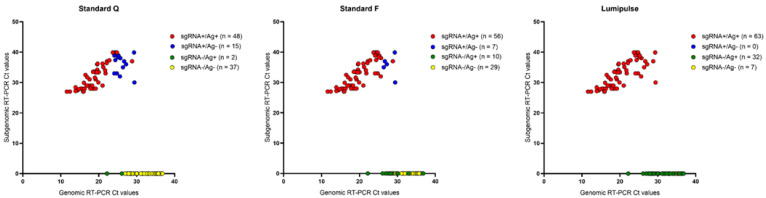
Distribution of NPS samples from 82 symptomatic COVID-19 patients tested for (reverse transcription-PCR) sgRNA and (Standard Q, Standard F, or Lumipulse) antigen detection according to number of days of symptoms before sample collection. Colors indicate groups of results stratified as sgRNA positive/Ag positive (red), sgRNA positive/Ag negative (blue), sgRNA negative/Ag positive (green), or sgRNA negative/Ag negative (yellow). Ag, antigen; COVID-19, coronavirus disease 2019; Ct, cycle threshold; NPS, nasopharyngeal swab; sgRNA, subgenomic RNA.

## Data Availability

The data presented in this study are available on request from the corresponding author.

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
