# Peer review of "SARS-CoV-2 Antigen Test Results to Infer Active or Non-Active Virus Replication Status in COVID-19 Patients"

_diagnostics, 2022, doi:10.3390/diagnostics12061338_

Round 1

Reviewer 1 Report

In this paper, the authors investigate the possibility of use nasopharyngeal swab for differentiation between active (potentially transmissible) and inactive (potentially non-transmitted) COVID patients using SARS-CoV-2 sgRNA antigen test. The results showed a good correlation, reinforcing the real potential use of this diagnostic method, which is of extremely important for distinguish the persons that should be isolated from the society.

The paper is well writing and structured; I only suggested the modification of both tables to fit in only one page.

Author Response

In this paper, the authors investigate the possibility of use nasopharyngeal swab for differentiation between active (potentially transmissible) and inactive (potentially non-transmitted) COVID patients using SARS-CoV-2 sgRNA antigen test. The results showed a good correlation, reinforcing the real potential use of this diagnostic method, which is of extremely important for distinguish the persons that should be isolated from the society.

The paper is well writing and structured; I only suggested the modification of both tables to fit in only one page.

Answer: I am very grateful to the reviewer for appreciating the manuscript and relative findings. Regarding the suggested modification on both Tables, I asked the Assistant Editor (Ms. Ilies) to make this modification because the Tables’ layout was unmodifiable by us.

Reviewer 2 Report

The authors sought to perform and evaluation linking the presence of SARS-CoV-2 sgRNA to diagnostic antigen positivity of 3 commercial antigen assays, two rapid antigen assays and one antigen assay requiring a laboratory.  These experiments were performed in an attempt to distinguish individuals carrying actively replicating SARS-CoV-2 from individuals shedding inactive SARS-CoV-2 RNA.  The authors found correlates between antigen positivity, sgRNA presence or absence and time to test from the development of symptoms.  The majority of individuals that were antigen/sgRNA positive were tested within the first week of symptoms while antigen/sgRNA negatives were tested late in the course of infection (16-30 days).  The results are interesting and for the most part the labtoratory methods are sound.  I do have some concerns that the authors should expand on before this can be published.

Major comments:  What is the sensitivity and specificity of these assays compared to standard diagnostic SARS-CoV-2 RT-PCR?  The authors report the sensitivity and PPV compared to culture, which is likely not the best comparison to use as both the antigen assays presented here as well as the sgRNA and diagnostic RT-PCR assays are all more sensitive than culture methodology are, the authors need to find a better way to present this.  The authors need to include a more detailed description of the study limitations, ie, relatively low sample size, low sample size for asymptomatic participants, the variability in assay performance within asymptomatic indivduals, the disagreement between the Standard Q and Standard F assays and the sgRNA assay at days 8-15 for both assays, particularly for the snRNA+/Ag- samples etc.  Additionally, this data largely suggests the highest utility for antigen assays is early in the onset of symptoms, which is not a particularly novel finding.

Author Response

The authors sought to perform and evaluation linking the presence of SARS-CoV-2 sgRNA to diagnostic antigen positivity of 3 commercial antigen assays, two rapid antigen assays and one antigen assay requiring a laboratory. These experiments were performed in an attempt to distinguish individuals carrying actively replicating SARS-CoV-2 from individuals shedding inactive SARS-CoV-2 RNA. The authors found correlates between antigen positivity, sgRNA presence or absence and time to test from the development of symptoms. The majority of individuals that were antigen/sgRNA positive were tested within the first week of symptoms while antigen/sgRNA negatives were tested late in the course of infection (16-30 days). The results are interesting and for the most part the laboratory methods are sound. I do have some concerns that the authors should expand on before this can be published.

Answer: I am very grateful to the reviewer for appreciating the manuscript and relative findings. Regarding the requested revision on the manuscript, I sought to expand it according to the relevant issues raised by the reviewer.

Major comments: What is the sensitivity and specificity of these assays compared to standard diagnostic SARS-CoV-2 RT-PCR? The authors report the sensitivity and PPV compared to culture, which is likely not the best comparison to use as both the antigen assays presented here as well as the sgRNA and diagnostic RT-PCR assays are all more sensitive than culture methodology are, the authors need to find a better way to present this.

Answer: I added a brief comment on the sensitivity and specificity of SARS-CoV-2 antigen assays compared to RT-PCR assays. To comply with the assumption that culture is a suboptimal way to detect replication-competent virus, we chose to compare antigen assays not with viral culture but with a more sensitive RT-PCR assay (i.e., the sgRNA detection assay). See Introduction (first paragraph) and Discussion (second paragraph) of the revised manuscript.

The authors need to include a more detailed description of the study limitations, i.e., relatively low sample size, low sample size for asymptomatic participants, the variability in assay performance within asymptomatic individuals, the disagreement between the Standard Q and Standard F assays and the sgRNA assay at days 8-15 for both assays, particularly for the snRNA+/Ag- samples etc. Additionally, this data largely suggests the highest utility for antigen assays is early in the onset of symptoms, which is not a particularly novel finding.

Answer: As suggested, I added a new paragraph before the closing sentence to consider all the study limitations underscored by the reviewer. See Discussion (last paragraph before conclusions) of the revised manuscript.